A multi-task pipeline with specialized streams for classification and segmentation of infection manifestations in COVID-19 scans

El-bana Shimaa 1
http://orcid.org/0000-0002-3941-0061 Al-Kabbany Ahmad 2 3 4 alkabbany@aast.edu alkabbany@ieee.org
Sharkas Maha 4
1 Alexandria Higher Institute of Engineering and Technology , Alexandria , Egypt
2 Intelligent Systems Lab, Arab Academy for Science, Technology, and Maritime Transport , Alexandria , Egypt
3 Department of Research and Development, VRapeutic , Cairo , Egypt
4 Department of Electronics and Communications Engineering, Arab Academy for Science, Technology, and Maritime Transport , Alexandria , Egypt
Ventura Sebastian
Electronic publication date: 2020 Oct 19
Publication date: 2020
Volume: 6
Electronic Location ID: e303
Received 2020 Jun 30; Accepted 2020 Sep 26
Copyright: © 2020 El-bana et al.
Copyright year: 2020
Copyright holder: El-bana et al.
License: This is an open access article distributed under the terms of the Creative Commons Attribution License, which permits unrestricted use, distribution, reproduction and adaptation in any medium and for any purpose provided that it is properly attributed. For attribution, the original author(s), title, publication source (PeerJ Computer Science) and either DOI or URL of the article must be cited.
License URL: https://creativecommons.org/licenses/by/4.0/

Keywords: COVID-19, Deeplab, Medical imaging, Pneumonia, Transfer learning, Multimodal learning

Funding: The authors received no funding for this work.

==============================
We are concerned with the challenge of coronavirus disease (COVID-19) detection in chest X-ray and Computed Tomography (CT) scans, and the classification and segmentation of related infection manifestations. Even though it is arguably not an established diagnostic tool, using machine learning-based analysis of COVID-19 medical scans has shown the potential to provide a preliminary digital second opinion. This can help in managing the current pandemic, and thus has been attracting significant research attention. In this research, we propose a multi-task pipeline that takes advantage of the growing advances in deep neural network models. In the first stage, we fine-tuned an Inception-v3 deep model for COVID-19 recognition using multi-modal learning, that is, using X-ray and CT scans. In addition to outperforming other deep models on the same task in the recent literature, with an attained accuracy of 99.4%, we also present comparative analysis for multi-modal learning against learning from X-ray scans alone. The second and the third stages of the proposed pipeline complement one another in dealing with different types of infection manifestations. The former features a convolutional neural network architecture for recognizing three types of manifestations, while the latter transfers learning from another knowledge domain, namely, pulmonary nodule segmentation in CT scans, to produce binary masks for segmenting the regions corresponding to these manifestations. Our proposed pipeline also features specialized streams in which multiple deep models are trained separately to segment specific types of infection manifestations, and we show the significant impact that this framework has on various performance metrics. We evaluate the proposed models on widely adopted datasets, and we demonstrate an increase of approximately 2.5% and 4.5% for dice coefficient and mean intersection-over-union (mIoU), respectively, while achieving 60% reduction in computational time, compared to the recent literature.

Introduction

The Severe Acute Respiratory Syndrome CoronaVirus 2 (SARS-CoV-2) of the Coronaviridae Study Group of the International Committee on Taxonomy of Viruses (2020) is a strain of Severe Acute Respiratory Syndrome-related CoronaVirus (SARS-CoV or SARSr-CoV). The latter is a species of coronaviruses, which are a group Ribonucleic Acid (RNA) viruses. SARS-CoV-2 causes an infectious respiratory disease that is known as the Coronavirus Disease 2019 (COVID-19), since it was first identified in December 2019, following a pneumonia outbreak Lai et al. (2020), Sharfstein, Becker & Mello (2020). The first human-to-human transmission was confirmed in January 2020 Chan et al. (2020), and the World Health Organization (WHO) declared a pandemic on the 11th of March 2020. Over three million confirmed cases to date, hundreds of thousands of deaths, and a severe socioeconomic impact in hundreds of countries that are hit by the virus (Li et al., 2020b; Wu, Leung & Leung, 2020) have induced significant efforts from governmental, public, and private sectors worldwide to manage the pandemic. One principal aspect of pandemic management and future epidemic prevention is the development of effective, efficient, and scale-able diagnostic tools.

There are several diagnostic tools that have been used, or currently under development, for SARS-CoV-2. To the best of our knowledge, nucleic acid tests are the most established and the most widely used tool to date (Tahamtan & Ardebili, 2020); in particular, the Polymerase Chain Reaction (PCR) and its variants, such as Quantitative PCR (qPCR) and Reverse Transcription PCR (RT-PCR). PCR is a DNA and RNA cloning technique that is used to amplify/augment DNA/RNA samples required in micro biology studies. Even though it is characterized by high sensitivity and specificity, in addition to rapid detection, it is prone to producing false negatives. In part, this is due to the localized nature of the sample acquisition process, mainly as nasal, throat, and nasopharyngeal swabs, that is, an active virus could be present elsewhere along the respiratory tract. There are also other limitations for PCR-based tests including universal availability, especially amidst shortage of supplies, slow turnaround times, depending on the resources of the lab, and in many cases, it is required to repeat the tests several times before they can be confirmed (Chu et al., 2020). Other diagnostic tools include antibody tests which can give an indication on whether a person was previously infected by the virus. However, they are still not well established; hence, they are not widely used. It is worth mentioning that the recent literature features recommendations for combining more than one diagnostic tool. Tahamtan & Ardebili (2020), for example, suggested the adoption of a combination of qRT-PCR and CT scans for robust management of COVID-19.

Using CT scans and other modalities, such as X-ray, falls under an ever-growing area of high-paced research, namely, medical imaging diagnostics. It has been emerging as a reliable disease diagnosis tool, with several recent research findings referring to a performance that is on-par with human performance (Liu et al., 2019; Shen et al., 2019). In a large part, this is due to the advances that are taking place in developing new machine learning techniques. This has resulted in the emergence of the Human-in-the-loop (HITL) AI framework (Patel et al., 2019), in order to harness the power of both approaches while avoiding their respective limitation simultaneously. For the current pandemic, though, using imaging as a first-line diagnostic tool for COVID-19 has been controversial to date (Hope et al., 2020; Fang et al., 2020; Zu et al., 2020; Ai et al., 2020). Meanwhile, to the best of our knowledge, there is a consensus on the possibility of using medical imaging as a digital second opinion, or a complement, to the gold standard PCR-based tests. For example, He et al. (2020) and Chen et al. (2020), respectively, highlighted CT scans as either a tool with comparable diagnostic performance as initial RT-PCR, or an important screening tool especially for patients who have initial negative results for the RT-PCR test. Accordingly, highly-paced research has been devoted to harness the potential of deep learning-based medical imaging diagnostics, towards the goal of providing a rapid, accurate, scale-able, and affordable diagnosis.

Deep neural network models have shown a considerable potential with regards to automatic detection of lung diseases (EL-Bana, Al-Kabbany & Sharkas, 2020; Polat & Danaei Mehr, 2019; Nasrullah et al., 2019). Thanks to their ability to extract and learn meaningful features, deep models can provide an effective alternative to manual labeling by radiologists—a task that is highly impacted by individual clinical experiences. Recent literature highlights the adoption of deep neural networks to analyze X-ray and CT scans, in order to recognize/classify COVID-19 from healthy subjects. Moreover, COVID-19 virus has a bilateral distribution of patchy shadows and ground glass opacity in early stages, which progress to multiple ground glass opacities and infiltrations in both lungs (Wang et al., 2020). These features are very similar to other types of pneumonia with only slight differences that are difficult to be distinguished by radiologists. Hence, deep models have been used to recognize/classify COVID-19 from other types of pneumonia, including bacterial and viral pneumonia (Narin, Kaya & Pamuk, 2020; Wang & Wong, 2020; Song et al., 2020). Deep models have also been used in the quantification and the segmentation of infection manifestations such as ground-glass opacity (GGO) and pulmonary consolidation, in early and late stages of infection, respectively (Ye et al., 2020; Ai et al., 2020).

In this research, we are inspired by a typical flow in a real-life scenario where a radiologist would employ a deep learning-empowered screening system, first, to recognize/diagnose COVID-19, then to quantify and segment infection manifestations in X-ray and CT scans. The development of multi-task pipelines has been the scope for previous research (Amyar, Modzelewski & Ruan, 2020). Nevertheless, we demonstrate either competitive or superior performance compared to the recent literature at every stage of the proposed pipeline. The following points summarize the principal contributions of this research:We outperformed the recent literature on COVID-19 recognition by attaining a classification accuracy of 99.4% for the two-class problem, that is (COVID-19/Non-COVID-19) and 98.1% for the four-class problem of recognizing COVID-19 among scans that involve normal cases, other types of pneumonia, in addition to COVID-19. To achieve this performance, we propose a training procedure that involves fine-tuning of an Inception-v3 architecture. We present the performance of this architecture under varying learning parameters, and using different performance metrics.

For the same stage, we show comparative analysis for learning using X-ray scans only against learning from X-ray and CT scans combined, that is, multi-modal learning, and we demonstrate a potential advantage for the latter.

We propose a CNN architecture for multi-label recognition/classification (ML-CNN) of different types of lung infection manifestations. Particularly, we solve the problem of identifying the probabilities of having infection manifestations, such as Ground Glass Opacities (GGO), Pleural Effusion (PE), and Consolidation, in medical scans. This is envisaged to have a potential role in the early diagnosis of COVID-19. It is worth mentioning that this problem was not addressed by previous work on multi-task pipelines for COVID-19 (Amyar, Modzelewski & Ruan, 2020).

We adapt knowledge from another domain, namely, pulmonary nodule detection, to enhance the segmentation of lung infections in chest CT scans. Particularly, we employ our own previous work (EL-Bana, Al-Kabbany & Sharkas, 2020) on improving semantic segmentation of pulmonary nodules using the recently proposed DeepLab-v3+ architecture. Moreover, using Xception network as a feature extractor backbone, we evaluate the performance of the DeepLab model, which suits client-side applications.

We propose a new learning procedure for semantic segmentation of infection manifestations. It involves the training of multiple streams, each of which is specialized to segment a specific type of manifestations. We demonstrate the effectiveness of this procedure over single stream-based segmentation, and compared to the recent literature, we attain an increase of approximately 4.5% and 2.5% for mean intersection-over-union (mIoU) and dice coefficient, respectively.

The rest of the article is organized as follows: Previous research that incorporates deep learning methods for COVID-19 diagnosis and infection segmentation is presented in “Related Work”. “Proposed Methods” discusses the proposed multi-stage pipeline, and we elaborate on the adopted datasets, data augmentation methods, and pre-processing techniques. “Results and Discussion” is dedicated to highlight and discuss the Experimental results, and finally the work is concluded in “Conclusion”.

Related work

This research intersects with four main areas in the literature, namely, COVID-19 recognition based on deep models, segmentation of COVID-19-related infection manifestations based on deep models, multi-task pipelines that have the ability to accomplish both tasks, and multi-stream recognition pipelines. In the rest of this section, we highlight the most relevant (to the proposed work) in these four areas.

The literature on COVID-19 diagnosis features end-to-end deep models as well as transfer learning approaches. Sethy & Behera (2020), for example, proposed a COVID-19 classification method in X-ray images using deep features that are computed using a pre-trained convolutional neural network (CNN) model, and an SVM classifier. This method attained an accuracy of 95.38% with the ResNet50 model employed as the feature extractor. In Li et al. (2020a), a retrospective, single-center, study was conducted on 78 patients. They aimed at investigating the correlation between CT-based manifestations and clinical classification of COVID-19. With an attained sensitivity of 82.6% and a specificity of 100.0%, they concluded that CT-based quantitative analysis is highly correlated with the clinical classification of COVID-19. They also pointed out that CT visual quantitative analysis is highly consistent in terms of the Total Severity Score that was introduced in their research. Ozkaya, Ozturk & Barstugan (2020) used a dataset of 150 CT scans to generate two sub-datasets of 16 × 16 and 32 × 32 patches. Deep features were then computed and an SVM classifier was trained on producing binary labels. They also proposed a novel method for feature ranking and fusion to enhance the performance of the proposed approach. An accuracy of 98.27% and 98.93% sensitivity were attained on the latter sub-dataset of patches. A weakly-supervised software system was developed in Zheng et al. (2020). It adopts deep learning and uses 3D CT volumes to detect COVID-19, achieving a specificity and sensitivity of 0.911 and 0.907, respectively.

The U-Net model is a CNN that was proposed by Ronneberger, Fischer & Brox (2015), and is among the widely-adopted neural networks in medical image segmentation. It was further extended to 3D U-Net (Çiçek et al., 2016), and UNet++ (Zhou et al., 2019) that showed promising performance on various image segmentation tasks. Zhou, Canu & Ruan (2020) proposed a U-Net-based segmentation technique that addressed COVID-19, and that employed an attention mechanism on 100 CT slices. They obtained a Dice Score of 69.1%. In our previous work (EL-Bana, Al-Kabbany & Sharkas, 2020), DeepLab-v3+ (Chen et al., 2017) was shown to outperform U-Net in pulmonary nodule segmentation. Fan et al. (2020) proposed a novel COVID-19 deep model for lung infection segmentation (Inf-Net) to identify infected regions from chest CT scans in an automated manner. On ground glass opacities and consolidation, Inf-Net achieved a dice coefficient of 0.646 and 0.238, respectively. Also, Elharrouss, Subramanian & Al-Maadeed (2020) proposed a deep-learning-based, multi-task, two-stage approach for infection segmentation in CT-scans. The first stage involves the possibly-infected lung regions, which is followed by segmenting the infections in these regions.

Amyar, Modzelewski & Ruan (2020) proposed a deep learning model that jointly identifies COVID-19 and segments the related lesions in chest CT scans. Their three-arm model consisted of a common encoder and two decoders for image reconstruction and segmentation, where the image reconstruction stage is meant to enhance feature representation. The third arm featured a multi-layer perceptron neural network for COVID-19 recognition, that is, a binary classification problem. For the segmentation task, they achieved a dice coefficient of 0.78. Wu et al. (2020) proposed a COVID-19 classification and segmentation system, that was trained on a dataset containing 144,167 CT scans, collected from 400 COVID-19 patients and 350 uninfected cases. Their JCS model achieved a 78.3% Dice Coefficient on the segmentation test set, and a sensitivity of 95.0%, and a specificity of 93.0% on the classification test set.

Deep networks with multiple streams have been employed in visual recognition applications. To the best of our knowledge, Simonyan & Zisserman (2014) were the first to adopt a two-stream ConvNet architecture, which incorporates spatial and temporal networks, for action recognition in videos. The proposed architecture involved training the second stream on optical flow, and it was shown to attain a very good performance despite limited data. Following the work of Simonyan & Zisserman (2014), other multi-stream techniques that adopt other modalities (Zhang et al., 2016) were proposed. In contrast to these previous techniques, our proposed multi-stream approach for segmenting infection manifestations trains each stream on a different label, rather than training each stream on a different modality of the whole dataset (all the labels). The latter is still a point of future research, though.

Proposed methods

A machine learning-empowered system for COVID-19 diagnostics inherently involves multiple tasks. As a digital second opinion for radiologists, the system would first be required to recognize COVID-19 in medical scans. It might further be asked to differentiate between COVID-19 and other types of pneumonia. Following the recognition of COVID-19, the system would be required to identify the probability of presence of different infection manifestations, and would further be asked to segment the regions corresponding to these manifestations accurately. Figure 1 depicts the proposed pipeline which realizes the aforementioned tasks. First, we employ the Inception-v3 model for image classification, particularly, for COVID-19 recognition. Second, we train a multi-label classifier to infer the probability of different types of infection manifestations, namely, Ground Glass Opacities (GGO), Pleural Effusion (PE), and Consolidation. The third stage involves feeding COVID-19 CT scans to DeepLab-v3+ model, which produces binary segmentation masks that highlight the regions corresponding to infection manifestations. To alleviate the impact of the limited amount of data, we use data augmentation techniques throughout the proposed pipeline. In the rest of this section, we elaborate on the datasets that are used for the training and testing of the proposed models, we elaborate on the adopted data augmentation techniques, and we discuss the implementation details of each of the three stages in the pipeline.

Figure 1 The block diagram of the proposed method.

Datasets

To the best of our knowledge, the research community still lacks a comprehensive dataset that involves CT and/or X-ray scans and that suits both diagnosis and segmentation tasks at the same time. This necessitates the reliance on multiple datasets if the goal is to develop a multi-task pipeline. For training the proposed deep models, we used the following datasets, which involve two of the most widely used datasets in the recent literature (Fan et al., 2020):COVID-19 CT Segmentation Dataset: This dataset includes 100 axial CT images from 40 patients with COVID-19. The images were segmented by a radiologist using Three labels: ground-glass, consolidation and pleural effusion. Figure 2 shows an example of CT COVID-19 slice from the dataset.

The COVID-19 Image Data Collection Repository on GitHub: This dataset is hared by Dr. Joseph Cohen. It is a growing collection of deidentified chest X-rays (CXRs) and CT scans from COVID-19 cases internationally (Cohen, Morrison & Dao, 2020).

The RSNA Pneumonia Detection Challenge Dataset: This dataset is available on Kaggle, and it contains several deidentified CXRs and includes a label indicating whether the image shows evidence of pneumonia. Figure 3 shows different examples of X-ray images from the dataset.

Figure 2 An example of a CT scan.

(A) COVID-19 CT axial slice, (B) ground truth segmented mask. The white regions in the latter represent the consolidation, while the dark gray regions represent pleural effusion, and light gray regions represent ground-glass opacities. Please see sub-section 1.

Figure 3 Examples of input X-ray images from the adopted datasets.

(A) Covid-19, (B) normal, (C) Pneumonia Bacteria, (D) Pneumonia Virus. Please see subsection 1.

Preprocessing and data augmentation

All medical scans were resized to have the shape of 512 × 512 × 3. The Contrast Limited Adaptive Histogram Equalization (CLAHE) method is used for enhancing small details, textures and local contrast of the images (Zuiderveld, 1994). Local details can therefore be enhanced even in the regions that are darker or lighter than most of the image (Koonsanit et al., 2017). To avoid over-fitting, since the number of CT volumes is limited, we applied data augmentation strategies such as random transformations. These transformations include rotation, horizontal and vertical translations, zooming and shearing. For each training sample, the transformation parameters were randomly generated and the augmentation was identically applied for each slice in the sampled image. We augmented each training sample 34 times, and each validation and test sample 51 times. The augmentation for the training, validation and testing datasets, rather than the training dataset only, is done in accordance with recent findings on the impact of test-time augmentation on the system performance (Perez et al., 2018). More details about the medical scans included in the adopted datasets are summarized in Table 1, such as the types of involved cases, the number of slices in each case and their modalities, and the total number of slices after augmentation.

Table 1 Details of the medical scans included in the adopted datasets, such as the cases available, the number of slices in each case and their modalities, the total number of slices after augmentation, and the task supported by each type of slices.

	Case	Modality	#Slices	Total after Augmentation	Task	
COVID-19	COVID-19	X-rays	60	1,995	Diagnosis	
	COVID-19 (with segmented masks)	CT Scan	100	3,724	Diagnosis + Segmentation	
Pneumonia	Bacterial Pneumonia	X-rays	70	2,122	Diagnosis	
	Viral Pneumonia	X-rays	70	2,277	Diagnosis	
Normal	Normal	X-rays	70	2,485	Diagnosis	

COVID-19 recognition using transfer learning on the inception-v3 architecture

The Inception-v3 architecture is an evolution of the GoogLeNet architecture. Prior to GoogLeNet, such as in the AlexNet and VGGNet architectures, a standard structure for CNNc consisted of stacked convolutional layers, max-pooling, and full-connected layers. To avoid over-fitting, computational demand, and exploding or vanishing gradients, the inception architecture encouraged sparsity through local sparse structures, namely, the Inception Modules/Blocks. Each of these blocks consists of four paths, and contains filters (convolutions) of different sizes, providing the ability to extract patterns at different spatial sizes. Convolutional layers that consist of 1 × 1 filters were used to make the network deeper, and to reduce the model’s complexity and the number of parameters, by reducing the number of input channels. The 1 × 1 convolutional layers also add more non-linearity by using ReLU after each 1 × 1 convolutional layer (Mahdianpari et al., 2018). The fully connected layer in this architecture is replaced with a global average pooling layer. In comparison to GoogLeNet, Incpetion-v2 featured the factorization of convolutions into smaller convolutions, while Incpetion-v3 extended Incpetion-v2 by batch-normalization of the fully connected layer of the auxiliary classifier (Szegedy et al., 2016). Figure 4 depicts a compressed view of the Inception-v3 (Xia, Xu & Nan, 2017) model.

Figure 4 A schematic diagram of the Inception-v3 architecture, inspired by the research in Szegedy et al. (2016).

In the first stage of the proposed pipeline, we fine-tuned an Incpetion-v3 architecture, which consists of a feature extraction stage, followed by a classification stage. Instead of training the whole architecture from scratch, we started from a model that is pre-trained on ImageNet. We left the weights of the pre-trained model untouched while the final layer is retrained from scratch. The number of classes in the dataset determines the number of output nodes in the last layer. In “Results and Discussion”, we discuss the impact of varying learning parameters, such as the number of steps and the learning rate, on the attained accuracy. We also demonstrate the performance of fine-tuning using multi-modal data, that is, X-rays and CT scans, as compared to fine-tuning using X-rays only.

A CNN architecture for multi-label recognition of infection manifestations in chest CT scans

There are several differences between the proposed pipeline and previous work on multi-task models for COVID-19 (Amyar, Modzelewski & Ruan, 2020). One principal difference, though, is that the second stage of our pipeline addresses a problem that was not handled by previously proposed models (Amyar, Modzelewski & Ruan, 2020), namely, the inference of the probabilities of presence of different infection manifestations, namely, Ground Glass Opacities (GGO), Pleural Effusion (PE), and Consolidation. Given that the output of the segmentation stage is a binary mask, important insights are missing with regards to the types of manifestations that correspond to the segmented regions, that is, the white regions in the output mask.

COVID-19 CT scans have featured three types of manifestations, namely, ground-glass opacity, consolidation and pleural effusion. Moreover, a scan may include one or more types of infections; hence it is a multi-label image recognition/classification problem. Towards the goal of recognizing different manifestations, we propose the CNN architecture that is shown in Fig. 5. The output of this architecture is a vector of three probabilities for the presence of ground-glass opacities, consolidations and pleural effusion in a CT scan. In a sense, the output of this stage complements the information obtained from binary segmentation masks, which will be addressed by the third stage of the pipeline. In addition, we envisage the second stage to have a significant role in early diagnosis even if the output from the first stage does not indicate signs for COVID-19.

Figure 5 The proposed CNN model for multi-label classification of infection manifestations.

As depicted in the figure, the output from the model are the probabilities of having different types of infection manifestations in chest CT scans.

The convolutional layers consist of M kernels of size N × N. Max-pooling is applied in non-overlapping windows of size 2 × 2. Every max-pooling reduces the size of each patch by half. Two dense layers with 128 and 64 neurons respectively are used with a dropout of 0.5 to avoid over-fitting, and the elu activation function is applied. The last layer is a dense layer for image classification using a sigmoid function to obtain the multi-label predictions and a cross entropy as the loss function. For N > 2, that is, multi-label classification, we calculate a separate loss per observation for each class label and sum the result as follows: (1) loss=−∑i=1Nyilog⁡(y^i)

where, N is the number of classes, y is the corrected label, y^ is a predicted output.

Another principal difference between the proposed model and the work in Amyar, Modzelewski & Ruan (2020) is that we deal with each task in the pipeline separately, that is, there is no common encoder. Hence, we are able to harness the power of different architectures in each task. This becomes apparent in the third stage where we adopted the DeepLab-v3+ model for segmentation, which was shown to achieve significantly better results (EL-Bana, Al-Kabbany & Sharkas, 2020) compared to U-NET that was adopted in Amyar, Modzelewski & Ruan (2020).

Segmenting infection manifestations with knowledge adaptation from pulmonary nodule segmentation

The third stage of the proposed pipeline uses the first dataset in sub-section 1, and is concerned with pixel-level segmentation of the regions corresponding to infection manifestations in CT scans. We capitalize on our previous research work in EL-Bana, Al-Kabbany & Sharkas (2020) in which we employed the DeepLab-v3+ model with CT scans to enhance the segmentation of pulmonary nodules, and in which we attained competitive results compared to the recent literature. The DeepLab-v3+ model was developed by Google, and it involves a simple decoder module to refine the segmentation masks along object boundaries. The model is fed with a single CT slice, and the corresponding ground truth mask showing the lesion locations is expected at the output. We explain the elements of the adopted model as follows:Atrous Separable Convolution: This form of convolution (Chen et al., 2019) is meant to reduce the complexity of the proposed model without compromising the performance. It is applied in the depth-wise convolution, where a depth-wise separable convolution replaces the standard convolution with two consecutive steps, namely, a depth-wise convolution followed by a point-wise convolution (i.e., 1 × 1 convolution). For 2D signals, each location i on the output feature map y, atrous convolution is computed as follows: (2) y[i]=∑k[xi+r.k]w[k],

where w is a convolution filter. The stride with the sampled input signal is determined by the atrous rate r. Standard convolution, though, is a particular case with r = 1.

Encoder: In segmentation tasks, objects in images as well as their locations represent the essential information required to accomplish successfully the computation of segmentation masks. This information is expected to get extracted by the encoder. In the proposed pipeline, the primary feature extractor in the DeepLab-v3+ model is an Aligned Xception model—a modified version of the Xception-65 model (Chollet, 2017). Xception is a modified version of the Inception module, in which Incpetion modules are replaced with separable depth convolutions. Moreover, in Aligned Xception, we use depthwise separable convolution with striding instead of all the maximum pooling operations. After each 3 × 3 depthwise convolution, extra batch normalization and ReLU activation are applied. Also, the depth of the model is increased without varying the entry flow of the network structure. Figure 6 depicts the modified Xception model.

Decoder: In this stage, the features computed during the encoding phase are employed to compute the segmentation masks. First, we bilinearly-upsample the encoder features by a factor of 4, before we concatenate them with the corresponding low-level features. 1 × 1 convolution is used on the low-level features before concatenation, in order to decrease the number of channels. After the concatenation, 3 × 3 convolutions are applied to enhance the features, which is followed by another bilinear upsampling by a factor of 4, as shown in the DeepLab-v3+ model in Fig. 1.

Figure 6 The modified Xception model (Chen et al., 2018) which is used as the backbone (feature extractor) for the DeepLab-v3+ model in the segmentation stage of the proposed pipeline.

In this work, we started from a pre-trained DeepLab-v3+ model. Particularly, we adapt another knowledge domain, namely, the pulmonary nodule segmentation, to enhance the segmentation of COVID-19 manifestations in CT-scans. We used the pre-trained model weights that were obtained in EL-Bana, Al-Kabbany & Sharkas (2020). Furthermore, since we focus on the enhancement of segmentation masks, we propose a new learning procedure that involves specialized streams, each of which features a DeepLab-v3+ model that trains to segment a specific type of manifestations. In the next section, we present the results of the proposed pipeline, and we elaborate on the gain of training multiple specialized streams as compared to a single-stream pipeline.

Results and Discussion

All the simulations were carried out on a machine with a GeForce GTX 1080Ti GPU, and 8 GB of VRAM. We used Python as the primary programing language and Tensorflow as the backbone in all the experiments. This research implements a new multi-task pipeline that is capable of accomplishing the following types of tasks: (1) COVID-19 classification in X-rays and CT scans, (2) Multi-label recognition of COVID-19 manifestations in CT scans, and (3) and segmentation of COVID-19 manifestations in CT scans. We adopted the most commonly used performance metrics in the respective areas, that is, for classification and segmentation, which are: sensitivity, specificity, accuracy, precision, F1-Score, Dice Coefficient (DSC), Intersection over Union (IoU), and Matthews Correlation Coefficient (MCC). The mathematical expressions for computing the aforementioned metrics are are given by: (3) Accuracy=TP+TNTP+TN+FP+FN,

(4) Senstivity=TPTP+FN,

(5) Specificity=TNTN+FP,

(6) Precision=TPTP+FP,

(7) MCC=(TP∗TN)−(FN∗FP)(TP+FN)∗(TN+FP)∗(TP+FP)∗(TN+FN),

(8) F1-Score=2∗precision.Recallprecision+precision,

(9) DSC=2|A⋂B||A|+|B|,

(10) IoU=TPTP+FP+FN,

where TP, FP, FN, and TN are the number of True Positives, False Positives, False negatives, and True Negatives, respectively.

For the segmentation task, our training set contains 5,000 COVID-19 images and the test set has 403 images. For the classification task, however, the training set contains the 9,618 images, and 955 images are included in the test set. For the two-class version of the classification problem, that is, COVID-19 vs. Normal, the total number of training images are 5,219, and 536 images are included in the test set. Train-test split was used for the evaluation in the three tasks. In the rest of this section, we refer to the COVID-19 classification of stage 1 as Task 1, to the multi-label recognition problem of stage 2 as Task 2, and the segmentation problem of stage 3 as Task 3.

Results of Task 1: classification using a fine-tuned inception-v3 model

For fine-tuning the Inception-v3 model, we used a batch size of 100 for 2,800 steps/iterations. Starting from a pre-trained model on ImageNet, we removed the weights of the last layer and re-trained it using X-ray and CT scans. For the four-class version of the recognition problem, that is, COVID-19, Normal, Viral Pneumonia, and Bacterial Pneumonia, the number of output nodes that is equal to the number of the classes is set to 4. For the two-class version of the recognition problem, that is, COVID-19 and Normal, the number of output nodes is set to 2. The last layer of the model was trained with the back-propagation algorithm, and the weight parameter is adjusted using the cross-entropy cost function by calculating the error between the softmax layer output and the specified sample class label vector.

Table 2 summarizes the results of the fine-tuned Inception-v3 model using 0.01 learning rate on the two-class and the four-class problems. After 2,800 steps for 4 classes, we achieved an accuracy of 99.9%, 97.71%, and 98.1% for the training, validation and testing, respectively. For 2 classes, however, we achieved an accuracy of 98.84%, 99.08%, and 99.4% for the training, validation and testing respectively. The confusion matrices of the two-class and four-class cases are shown in Figs. 7A and 7B respectively. We also show the variations of the accuracy and cross-entropy for that model for classification of 2 classes in Fig. 8.

Table 2 Classification results of the fine-tuned Inception-v3 model for the two-class and the four-class COVID-19 recognition problems.

Please see text for more details.

# of Classes	Training accuracy (%)	Validation accuracy (%)	Test accuracy (%)	Training cross entropy	Validation cross entropy	
Classes	98.59	97.71	98.1	0.07687	0.09425	
Classes	99.84	99.08	99.4	0.01626	0.03283	

Figure 7 Confusion matrices of the fine-tuned Inception-v3 model for the two-class and the four-class COVID-19 recognition problems.

(A) Confusion matrix for two classes, (B) Confusion matrix for four classes.

Figure 8 The variation of accuracy and cross-entropy using the Inception-v3 model with 2-classes X-ray dataset.

(A) The variation of accuracy. (B) The variation of cross-entropy.

We also compared the performance of the adopted model with other models in the recent literature. Table 3 presents a summary of the accuracy, sensitivity, specificity, precision, F1-Score and MCC attained by different architectures. We demonstrate that the transfer learning approach with Inception-v3 surpassed all other architectures by achieving a 99.4% accuracy in case the training was done using X-rays only. We further tried to train using multi-modal data, that is, using X-rays and CT scans, and we achieved a 99.5% accuracy. We argue that the increase in the attained accuracy, using multi-modal data, is due to the 3D cues that are provided by, and inherently exist, in CT scans, but are missing in X-rays. It is worth mentioning that in order to avoid imbalanced data, we made sure that we have an equal number of X-rays and CT scans when we trained with multi-modal data. Particularly, we under-sampled the X-rays so that we get a number equal to the number of available CT scans. The under-sampling was done randomly, and we report the results that corresponds to the average of 5 runs. Complete results for each of the 5 runs are given in Table 4. It is worth mentioning that due to the limited number of available CT scans, uni-modal learning (using X-rays only) was carried out using a larger number of scans, yet multi-modal learning attained a slightly higher accuracy—99.4% for the former vs. 99.5% for the latter. We report this comparison to highlight that multi-modal learning is worth further exploration when larger number of CT scans becomes available.

Table 3 Comparing the recognition performance of the proposed model with other models in the recent literature.

	Method	Modality	Accuracy (%)	Senstivity (%)	Specificty (%)	Precision (%)	F1-score (%)	FPR	MCC (%)	
4-Classes	Alexnet (Loey, Smarandache & Khalifa, 2020)	X-ray	66.67	66.67	–	64.68	65.66	–	–	
	Resnet18 (Loey, Smarandache & Khalifa, 2020)	X-ray	69.46	66.67	–	72.50	69.46	–	–	
	ShuffleNet + SVM (Sethy & Behera, 2020)	X-ray	70.66	65.26	–	–	58.79	17.36	–	
	Googlenet (Loey, Smarandache & Khalifa, 2020)	X-ray	80.56	80.56	–	84.17	82.32	–	–	
	CNN (Zhao et al., 2020)	CT	84.7	76.2	–	97.0	85.3	–	–	
	Inception-v3 + SVM (Sethy & Behera, 2020)	X-ray	96.00	90.26	–	–	90.28	4.86	–	
	DenseNet201 + SVM (Sethy & Behera, 2020)	X-ray	97.33	93.86	–	–	93.86	3.06	–	
	XceptionNet + SVM (Sethy & Behera, 2020)	X-ray	97.33	93.00	–	–	93.00	3.50	–	
	VGG-16 + SVM (Sethy & Behera, 2020)	X-ray	97.33	94.20	–	–	94.20	2.90	–	
	InceptionResnetV2 + SVM (Sethy & Behera, 2020)	X-ray	97.33	91.13	–	–	91.74	4.43	–	
	Ours TL-Incep-V3	X-ray	98.1	98.02	98.03	98.2	98.2	2	–	
2-Classes										
	DRE-Net (Song et al., 2020)	CT	64	92	96.12	96	94	3.85	88.3	
	DenseNet	CT	96.25	96.29	96.21	96.29	96.29	–	–	
	VGG-16	CT	96.93	99.20	94.67	94.90	97.0	5.33	93.96	
	Resnet-50	CT	97.33	99.20	95.47	95.63	97.38	4.53	94.73	
	GoogleNet	CT	97.87	96.93	98.80	98.78	97.85	1.2	95.75	
	Ozkaya, Ozturk & Barstugan (2020)	CT	98.27	98.93	97.60	97.63	98.28	2.4	96.54	
	MobileNet v2	X-ray	97.40	99.10	97.09	–	–	–	–	
	VGG19	X-ray	98.75	92.85	98.75	–	–	–	–	
	Ours TL-Incep-V3	X-ray	99.4	99.5	99.1	99.1	99.3	0.9	98.7	
	Ours TL-Incep-V3	CT + X-ray	99.5	99.8	98.2	99.2	99.5	0.81	99.0	

Table 4 The prediction performance for the five runs which were carried out on two-class, multi-modal data (X-ray and CT scans).

Test no	Accuracy (%)	Senstivity (%)	Specificity (%)	Precision (%)	F1-Score (%)	FPR	MCC	
Test-1	99.7	100	99.4	99.5	99.7	0.57	99.4	
Test-2	99.7	100	99.4	99.5	99.7	0.56	99.5	
Test-3	99.4	100	98.9	98.9	99.4	1.14	98.9	
Test-4	99.7	100	99.4	99.4	99.7	0.64	99.4	
Test-5	99.1	99.4	98.9	98.8	99.1	1.16	98.2	
Mean	99.5	99.8	99.2	99.2	99.5	0.81	99.0	

Results of Task 2: multi-label classification of infection manifestations in CT scans

In the multi-label classifier, each convolutional layer is followed by maxpooling and dropout regularization of 25% to prevent the model from over-fitting. We used 5 × 5 filter for convolution and 2 × 2 for maxpooling, then, a flattening operation is carried out for classification. The activation function is elu for all the layers, except for the last one which is a sigmoid, in order to generate a probability for each label—ground glass, consolidation, and pleural effusion. The loss function is the binary cross-entropy and the metric is the accuracy, with Adam as the optimizer (Kingma & Ba, 2014). The model was trained for 50 epochs. Figure 9 shows the confusion matrix for the three labels in the COVID-19 dataset. More performance metrics are given in Table 5. It is worth mentioning that we do not report a comparison between our performance at this stage and the recent literature. This is because, to the best of our knowledge, this research is the first to address the problem of recognizing different types of infection manifestations. Even for the recently proposed multi-task model in Amyar, Modzelewski & Ruan (2020), its recognition arm addressed binary classification, which is identical to the two-class problem addressed by stage 1 of our pipeline. The segmentation stage in Amyar, Modzelewski & Ruan (2020) did not address multi-label infection recognition either, as it was limited to produce binary masks.

Figure 9 The confusion matrix of the adopted multi-label classifier.

Table 5 Different performance metrics for the adopted multi-label classifier.

We show the performance for individual labels as well as the overall performance.

Class name	Accuracy (%)	Precision (%)	Senstivity (%)	F1-Score (%)	
Pleural effusion	91.31	83	90	86	
Ground glass	89.46	91	80	85	
Consolidation	93.72	88	93	90	
Overall	87.2	87.3	87.6	87	

Results of Task 3: semantic segmentation of COVID-19 infection manifestations using multiple specialized streams

As mentioned in the previous section, we initialized the DeepLab-v3+ model using the weights of the checkpoint used to segment the lung cancer nodules in our previous work (EL-Bana, Al-Kabbany & Sharkas, 2020). We set the learning rate to 0.0001, the momentum to 0.9, the weight decay to 0.00004, and the steps to 50,000. We also adjusted the atrous rates as [6, 12, 18] with an output stride of 16. In Fig. 10, we present the output segmentation masks on the COVID-19 validation set. The figure shows the segmentation output of each of the specialized streams, and the output of the all-class stream, that is, the single stream that was trained to segment all the classes of manifestations at the same time. To support subjective results with objective measures, we report in Table 6 the dice coefficient (DSC) and the mean Intersection over Union (IoU) attained by the all-class stream, each of the three specialized streams, and their average. Considering the performance of the specialized streams, which outperformed the single stream approach, we believe that this defines an accuracy-complexity trade-off, that is, in order to attain better DSC and IoU, the system needs to include multiple specialized streams. We also believe that given the COVID-19 pandemic management as an application, in which significant resources have already been invested, there is a higher priority for developing highly accurate systems over low-complexity systems.

Figure 10 The output segmentation masks of the adopted deep models.

The images in column 1 from (A) to (C) show the chest CT images of three scans. Column 2 from (D) to (F) shows the ground-truth masks for these three scans, where the white represents the consolidation, dark gray represents pleural effusion and light gray corresponds to ground-glass opacities. Column 3 from (G) to (I) depicts the segmentation results generated by our model for all classes where the red represents the consolidation, the green represents the pleural effusion, and the yellow represents the ground-glass opacities. The images in columns 3, 4, and 6 from (J) to (R) represent the output from the specialized stream that are trained to segment ground-glass opacities, pleural effusion, and the consolidation, respectively.

Table 6 A comparison between the performance of each of the specialized streams as well as the all-class stream, with regards to dice coefficient (DSC) and mean Intersection over Union (mIoU).

For all the streams, a DeepLab-v3+ model, with an Xception 65 as a feature extractor, is used.

	All-Class	Stream 1: PE	Stream 2: GGO	Stream 3: Consolidation	Multi-Stream Average	
DSC (%)	86.04	91.34	90.2	91.5	91.01	
mIOU (%)	75.5	84.06	82.15	84.46	83.5	

To compare the performance of the proposed approach with other models, we report the results for specific types of infection manifestations as well as the overall performance for all types of manifestations. Table 7 shows a manifestation-specific comparison between the performance of our model, namely, DeepLab-v3+ model with transfer learning from pulmonary nodule detection, and other models from the recent literature including previous research that adopted DeepLab-v3+. The comparison highlights the superiority of our approach consistently for the two types of manifestations. This represents approximately 41% and 290% increase in DCS of ground-glass opacities and consolidation, respectively, compared to the recent literature. For mIoU, the comparison yields an increase of approximately 77% and 500% in DCS of ground-glass opacities and consolidation, respectively.

Table 7 A quantitative comparison of manifestation-specific DSC and mIoU, for Ground-Glass Opacity and Consolidation, between our segmented method and other methods in the recent literature.

	Method	DSC	mIOU	
Ground-glass opacities	DeepLabV3+ (stride = 8) (Fan et al., 2020)	0.375	0.230	
	DeepLabV3+ (stride = 16) (Fan et al., 2020)	0.443	0.284	
	FCN8s (Fan et al., 2020)	0.471	0.308	
	Semi-Inf-Net+FCN8s (Fan et al., 2020)	0.646	0.477	
	Ours (DeepLab-v3+ + exception-65)	0.902	0.8215	
Consolidation	DeepLabV3+ (stride = 8) (Fan et al., 2020)	0.117	0.062	
	DeepLabV3+ (stride = 16) (Fan et al., 2020)	0.188	0.103	
	FCN8s (Fan et al., 2020)	0.221	0.124	
	Semi-Inf-Net+FCN8s (Fan et al., 2020)	0.238	0.135	
	Ours (DeepLab-v3+ + exception-65)	0.915	0.8446	

We further make a comparison that is not manifestation-specific, between the performance of the proposed approach and the recent literature. In Table 8, we demonstrate an increase of approximately 4.5% and 2.5% for mean intersection-over-union (mIoU) and dice coefficient, respectively, compared to the recent literature. Figure 11 depicts a subjective comparison using examples for the output segmentation masks on the COVID-19 validation set obtained using U-net (Chen, Yao & Zhang, 2020) and DeepLab-v3+ (ours). We also demonstrate less computational cost than the traditional test, the RT-PCR, and other diagnostic tools (Huang et al., 2020; Wu et al., 2020). We report this comparison in Table 9, which shows a 60% reduction in diagnosis/computational time per case. Table 10 summarizes the model hyper-parameters used in the three tasks that are accomplished by the proposed system.

Figure 11 Segmentation output visualization results.

(A) and (B) chest CT images of two scans. (C) and (D) ground-truth masks for these two scans, where the white represents the consolidation, dark gray represents pleural effusion and light gray corresponds to ground-glass opacities. (E) and (F) the outputs of the U-Net. (G) and (H) the segmentation results generated by our model.

Table 8 A quantitative comparison on the COVID-19 segmentation dataset between our segmentation method and other methods in the recent literature.

The comparison considers DSC and mIoU. It also considers the overall performance on the three different types of infection manifestations, that is, it is not a manifestation-specific comparison.

Method	DSC (%)	mIOU (%)	
U-Net + DL (Zhou, Canu & Ruan, 2020)	61.0	43.88	
U-Net + FTL (Zhou, Canu & Ruan, 2020)	66.7	50.15	
U-NET 512 × 512 (Amyar, Modzelewski & Ruan, 2020)	67.14	50.53	
AU-Net + DL (Zhou, Canu & Ruan, 2020)	68.5	52.09	
U-NET 256 × 256 (Amyar, Modzelewski & Ruan, 2020)	69.09	52.77	
AU-Net + FTL (Zhou, Canu & Ruan, 2020)	69.1	52.78	
Backbone + PPD + RA + EA (Fan et al., 2020)	73.9	58.6	
JCS (Wu et al., 2020)	77.5	65.2	
JCS‘ (Wu et al., 2020)	78.3	66.5	
Amine (Amyar, Modzelewski & Ruan, 2020)	78.52	64.6	
U-net (Chen, Yao & Zhang, 2020)	82	69.49	
M–A (Chen, Yao & Zhang, 2020)	85	73.91	
M–R (Chen, Yao & Zhang, 2020)	84	72.41	
Ours method	86.04	75.5	

Table 9 A comparison between the proposed method and other diagnostic tools in the literature concerning the average diagnosis time per case.

Method	(Won et al., 2018)	(Huang et al., 2020)	(Wu et al., 2020)	Ours	
Test Time	4 h	21.5 min	19 s	5.33 s	

Table 10 A summary of hyper-parameters used in the proposed model.

Task_no	Steps/Epochs	Learning rate	Optimizer	Momentum	Dropout	Weight decay	Batch size	
Task 1	2,800 steps	0.01	Gradient descent	–	–	–	100	
Task 2	50 epochs	0.01	Adam	–	0.5	–	64	
Task 3	50 K steps	0.0001	SGD	0.9	–	0.00004	8	

Conclusion

In this research, we proposed a multi-task pipeline for the recognition of COVID-19, and the classification and segmentation of related infection manifestations in medical scans. We are inspired by the emerging role that medical imaging-based diagnostics can play as a digital second opinion to manage the current pandemic. The proposed pipeline starts with a COVID-19 recognition stage. Towards this goal, we fine-tuned and Inception-v3 model which was pre-trained on ImageNet. We evaluated the performance of this model on two tasks, namely, the two-class problem of COVID-19/non-COVID-19 recognition, and the four-class problem of recognizing COVID-19 scans from other scans that correspond to normal, viral pneumonia, and bacterial pneumonia cases. We outperformed other techniques in the recent literature, consistently in both types of classification problems. To the best of our knowledge, we are also the first to highlight a potential advantage for multi-modal learning, that is, learning from X-rays and CT scans over learning from X-rays only. In the second stage, we addressed a problem that was not been addressed by the recent literature, namely, the identification of the probabilities of presence for different types of infection manifestations in medical scans. This stage was implemented using a multi-label CNN classifier, and we envisage its potential to serve in early detection of infection manifestations. It also complements the third stage which addresses the problem of computing binary masks for segmenting the regions corresponding to infection regions in CT scans. For effective segmentation, we adapted the knowledge from another domain, namely, pulmonary nodule segmentation. This approach resulted in an increase of approximately 2.5% and 4.5% for dice coefficient and mean intersection-over-union (mIoU), respectively, while requiring 60% less computational time, compared to the recent literature. All the stages of the proposed pipeline were trained and tested using widely adopted datasets, and evaluated using various objective measures. We also used data augmentation techniques to avoid over-fitting that might have occurred due to the relatively limited volume of available data. For further enhancement of the performance of the segmentation stage, we showed that using multiple streams can significantly improve the quality of the output masks, as measured by the DSC and mIoU, such that each stream is trained to segment a specific type of infection manifestations.

Additional Information and Declarations

Competing Interests

Author Contributions

Data Availability

Shimaa El-bana and Maha Sharkas declare that they have no competing interests.

Ahmad Al-Kabbany is the Founder of VRapeutic, and is currently serving as a member of the Research & Development Department's team.

Shimaa El-bana conceived and designed the experiments, performed the experiments, analyzed the data, performed the computation work, prepared figures and/or tables, authored or reviewed drafts of the paper, and approved the final draft.

Ahmad Al-Kabbany conceived and designed the experiments, performed the experiments, analyzed the data, authored or reviewed drafts of the paper, and approved the final draft.

Maha Sharkas conceived and designed the experiments, authored or reviewed drafts of the paper, and approved the final draft.

The following information was supplied regarding data availability:

Dataset and scripts are available at GitHub:

https://github.com/shimaaelbana/Classification-and-Segmentation-of-infection-manifestations-in-COVID-19-scans.

The datasets acquired for analysis are available at:

1. COVID-19 CT Segmentation Dataset: http://medicalsegmentation.com/COVID19/?fbclid=IwAR2uZZ9f0mBHHVF10hMhS0OQ82LMOxQ0YdMXBOc5lKKZYL1h1eNVFU_fUp.

2. The COVID-19 Image Data Collection Repository on GitHub: https://github.com/ieee8023/COVID-chestxray-dataset.

3. The RSNA Pneumonia Detection Challenge Dataset: https://www.kaggle.com/c/rsna-pneumonia-detection-challenge.

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
