# Peer review of "A multi-task pipeline with specialized streams for classification and segmentation of infection manifestations in COVID-19 scans"

_PeerJ Computer Science, doi:10.7717/peerj-cs.303_

## Round 0.1 · original submission · Major Revisions

One of the reviewers has serious concerns that have to be addressed before the paper was ready for publication. Please prepare a new version considering all their suggestions.

Reviewer 1 ·

Basic reporting

The goal of the work is to proposes a multi-task pipeline that takes advantage of the growing advances in deep neural network models. The authors used Inception-v3 and transfer learning, as well as widely adopted datasets. The paper is in the scope of PeerJ Computer Science. The introduction is extensive literature supported. Some grammatical errors are there in the content, need to be removed. Some images are low quality, such as Figure 1, 4, 5, 6, 7, 8, 9; authors should consider using vector graphic images such as EPS.

Experimental design

A big computational effort was made. The authors evaluated the segmentation proposal using COVID-19 CT Segmentation Dataset, which is suitable.

Random transformations such as rotation, horizontal and vertical translations, zooming and shearing, were applied to increase the training data. However, there is a big concern regarding data augmentation. The authors should describe in detail in their work how was data-augmentation applied, e.g. how many images were increased in each category? Was data augmentation used in train, validation and test? The above can affects the results if augmented images from training were used in validation or test.

Apparently, the authors used hold-out for training and testing respectively, however, this is one of the simplest evaluation methods considering the low number of available images; there are other more exhaustive evaluation methods such as k-fold cross validation and leave-one-out cross validation which are more appropriate in this context.

Validity of the findings

The authors should support the Conclusions based on the performance in the test data, in this regard, the experimental study needs to be clarified in order to sustain the validity of the work.

Reviewer 2 ·

Basic reporting

The present work proposes COVID19 detection and CT scan segmentation method using deep learning model for. The field is important, and the use of deep learning can help for quick diagnostic for COVID19 diagnostics.

Experimental design

The following comment can enhance the work:
1- Many methods use transfer learning for detection. The authors should give a simple comparison with the existing methods.
2- A table that summarize the hyperparameters and some details about the proposed segmentation model.
3- It better to move section dataset to be after the proposed method or as subsection of experiments section.
4- There are few methods for COVID19 lung infection segmentation. the authors should add these methods in comparisons including [1].

[1] Elharrouss, O., Subramanian, N. and Al-Maadeed, S., 2020. An encoder-decoder-based method for COVID-19 lung infection segmentation. arXiv preprint arXiv:2007.00861.

Validity of the findings

5- The quality of some figures should be enhanced. Like figure 8.

---

## Round 0.2 · accepted · Accept

After checking the comments of the reviewers, I think the paper can be accepted for publication. Nevertheless, I encourage authors to include the advice given by reviewer 1 in the definite version of the article.

Reviewer 1 ·

Basic reporting

The authors have added and clarified questions from previous revision. The introduction and methods are well literature supported. There are some grammatical errors in the manuscript, e.g. lines 242-246.

Experimental design

The authors used several evaluation metrics, some of them specially applied for imbalance data, which was suitable. In lines 324-329 the authors stated that the proposal was evaluated by using the following setting: Task 1 5000 for training and 403 for testing, Task 2 9618 for training and 955 for testing, Task 3 5219 for training and 536 for testing. Could you briefly clarify in this section of the manuscript how many times did you repeat this evaluation? For example, you evaluated Task 1 with 5 repetitions, choosing a set of 955 different images in each of them for testing, and 9618 for training. The header of Table 4 seems to give this information; however, it is expected full setting is explained in the section already mentioned.

Validity of the findings

Results corroborated the proposal outperformed other state-of-the-art techniques.

Reviewer 2 ·

Basic reporting

the authors respond to my comments

Experimental design

the presentation has been reviewed and the authors respond to my comments.

Validity of the findings

--

Additional comments

--

---

## Author Rebuttal · Round 0.2

**Arab Academy for Science, Technology & Maritime Transport**

Department of Electronics and Communications Engineering
Faculty of Engineering and Technology
Arab Academy for Science, Technology, and Maritime Transport
Abu-Qir, Alexandria, 21937
Egypt

*September 5, 2020*

Dear Editors,

We would like to express our gratitude for the reviewers' insightful comments on the manuscript.

We have responded to the comments of the respected reviwers in the attached rebuttal letter, and we have added and/or clarified the corresponding parts of the article accordignly.

We sincerely hope that the manuscript is now up to the quality required to get published in PeerJ.

Dr. Ahmad Al-Kabbany
Assistant Professor of Electronics and Communications Engineering
LinkedIn | ResearchGate
Founding Member, Intelligent Systems Lab
Co-founder and CEO, VRapeutic

*On behalf of the authors.*

# Reviewer 1

## Basic reporting

*The goal of the work is to proposes a multi-task pipeline that takes advantage of the growing advances in deep neural network models. The authors used Inception-v3 and transfer learning, as well as widely adopted datasets. The paper is in the scope of PeerJ Computer Science. The introduction is extensive literature supported. Some grammatical errors are there in the content, need to be removed. Some images are low quality, such as Figure 1, 4, 5, 6, 7, 8, 9; authors should consider using vector graphic images such as EPS.*

**Answer**: We would like to inform the respected reviewer that all the figures mentioned above, which were in PNG format, have been regenerated in PDF format. We hope that the figures in the current format are of sufficient quality. We have also reviewed the article for grammatical errors. If the reviewer still sees some errors, we would be glad to correct them.

## Experimental design

*A big computational effort was made. The authors evaluated the segmentation proposal using COVID-19 CT Segmentation Dataset, which is suitable.*

*Random transformations such as rotation, horizontal and vertical translations, zooming and shearing, were applied to increase the training data. However, there is a big concern regarding data augmentation. The authors should describe in detail in their work how was data-augmentation applied, e.g. how many images were increased in each category? Was data augmentation used in train, validation and test? The above can affects the results if augmented images from training were used in validation or test.*

*Apparently, the authors used hold-out for training and testing respectively, however, this is one of the simplest evaluation methods considering the low number of available images; there are other more exhaustive evaluation methods such as k-fold cross validation and leave-one-out cross validation which are more appropriate in this context.*

**Answer**: First, we would like to express our gratitude to the respected reviewer for the insightful comment regarding augmentation. This comment was the reason to correct a mistake in the simulation.

While the augmentation in task 1 (classification) and task 2 (multi-label classification) was done after the train-test split, we found that the augmentation in task 3 was done before splitting (by mistake). This **might have resulted** in what the respected reviewer mentioned, i.e., augmented images **might have been** used in testing. We have corrected the simulation, and we included the new results for task 3 in the updated version of the article. As for the details of the augmentation process, we have added more details on the number of augmentations per sample in lines 222-227. We also want to confirm that in line 223 in the article, we indicated more details regarding the augmentation process including the types of used transformations and how the transformation parameters were generated (randomly). Moreover, we want to confirm that the number of slices

before and after augmentation are indicated in Table 1. We have also indicated that the augmentation was **on each sample**. We are ready to provide further information, but we would be thankful if the respected reviewer can elaborate on the required information.

As for hold-out vs. cross-validation, we believe that the respected reviewer meant the results in Table 4. If we are correct, so we need to elaborate on an important piece of information. Table 4 is concerned with the results of multi-modal learning. Basically, we wanted to explore the impact of learning from CT and X-ray images together, instead of learning from only one modality of them. This is one of the contributions of the article since we have not seen it before, to the best of our knowledge, in the recent literature. That being said, we had a challenge in the unbalanced nature of the available data, i.e., the available X-ray images are much more than the available CT scans. To construct a "mixed balanced sample" of X-ray and CT images, we included the CT images and then chose an equal number of X-ray images randomly from all the available images. Even though we run hold-out (or alternatively, train-test split) on this sample, we constructed five different "mixed samples", each of which consists of the same set of CT images + randomly selected X-ray images of equal number to the CT images, in order to see the impact of training with a wider variety of data points. Going back to the comment of the respected reviewer, why didn't we run a k-fold, for example, on one sample? In our humble opinion, we have adopted a remarkably close approach. If, for example, we adopted a 5-fold approach, we would have trained our model on 5 train-test splits, right? In our propose approach, we also trained our model on 5 different train-test splits, and the results of the 5 splits were quite close. So, we argue that our results demonstrated model stability, even though it is not the "canonical k-fold". We appreciate any further guidance from the respected reviewer.

**Validity of the findings**

*The authors should support the Conclusions based on the performance in the test data, in this regard, the experimental study needs to be clarified in order to sustain the validity of the work.*

**Answer**: Based on the clarifications we presented above; we hope that the experimental study is now clearer for the respected reviewer. We have also reviewed the Conclusion section, and we made sure that each statement is justified by analysis in the Results section. We would be glad to omit any unjustified claim from the Conclusion and/or better clarify any existing statement.

**Basic reporting**

The present work proposes COVID19 detection and CT scan segmentation method using deep learning model for. The field is important, and the use of deep learning can help for quick diagnostic for COVID19 diagnostics.

**Experimental design**

The following comment can enhance the work:

1- _Many methods use transfer learning for detection. The authors should give a simple comparison with the existing methods._

**Answer**: As for task 1, as per the comment of the respected reviewer, we have added more comparisons with transfer learning-based methods in Table 3. As for task 3, we would like to highlight that the main difference with other methods is that we adopt an in-domain transfer learning approach from the area of pulmonary nodule detection. Particularly, we capitalize on the superiority of our own previous work in pulmonary nodule detection in order to benefit the research in another lung-related disease, namely, COVID19. We indicated this in lines 117-121. We are ready to add more information upon further instructions from the respected reviewer.

2- _A table that summarize the hyperparameters and some details about the proposed segmentation model._

**Answer**: A table summarizing the hyper-parameters is added as Table 6 in the updated version of the article.

3- _It better to move section dataset to be after the proposed method or as subsection of experiments section._

**Answer**: We highly value the suggestion of the respected reviewer. Meanwhile, we believe that we should introduce the adopted datasets before the proposed method for clearer presentation. Accordingly, moving the dataset section to after the proposed method or in the experiments section would not serve our presentation the way we hope, in our humble opinion. We appreciate the kind understanding of the respected reviewer.

4- _There are few methods for COVID19 lung infection segmentation. the authors should add these methods in comparisons including [1]._
_[1] Elharrouss, O., Subramanian, N. and Al-Maadeed, S., 2020. An encoder-decoder-based method for COVID-19 lung infection segmentation. arXiv preprint arXiv:2007.00861._

**Answer**: We are confirming that we have added the reference suggested by the respected reviewer in lines 162-166.

**Validity of the findings**

5- *The quality of some figures should be enhanced. Like figure 8*.

**Answer**: We would like to inform the respected reviewer that figures 1, 4, 5, 6, 7, 8, 9, which were in PNG format, have been regenerated in PDF format. We hope that the figures in the current format are of sufficient quality.